# Phytotoxic Activity Analysis of 2-Methoxyphenol and 2,6-Di-*tert*-butyl-4-methylphenol Present in *Cistus ladanifer* L. Essential Oil

**DOI:** 10.3390/plants14010022

**Published:** 2024-12-25

**Authors:** Diego Orellana Dávila, David F. Frazão, Amélia M. Silva, Teresa Sosa Díaz

**Affiliations:** 1Department of Plant Biology, Ecology and Earth Sciences, Faculty of Science, University of Extremadura, 06006 Badajoz, Spain; diorellan@alumnos.unex.es; 2Centro de Recursos Naturales, Medio Ambiente y Sociedad, Instituto Politécnico de Castelo Branco, 6001-909 Castelo Branco, Portugal; davidmfrazao@gmail.com; 3Department of Biology and Environment, School of Life Sciences and Environment, University of Trás os Montes e Alto Douro, Quinta de 20 Prados, 5001-801 Vila Real, Portugal; amsilva@utad.pt

**Keywords:** phytotoxicity, phenolic compounds, allelopathy, germination, seedling growth, bioherbicides, essential oil

## Abstract

The evaluation of the wide variety of allelochemicals present in allelopathic plants allows the detection of safer bioherbicides with new mechanisms of action. This study tested two phenolic compounds of *Cistus ladanifer* essential oil (2-Methoxyphenol and 2,6-Di-*tert*-butyl-4-methylphenol), which are commercially available. At 0.01 mM, these compounds, both separately and in combination (1/1), inhibited up to over 50% of germination, cotyledon emergence and seedling growth of *Lactuca sativa* for the tests conducted on paper. Against *Allium cepa*, cotyledon emergence and seedling growth were inhibited at 0.5 mM. When the tests were carried out in the soil, the mixture of the two study compounds significantly inhibited the germination of *L. sativa* and *A. cepa* when applied at 0.5 and 1 mM, respectively, and seedling growth inhibition was greater for the latter in the paper tests. The greatest inhibitions were observed, with the highest concentrations analysed. Although there was no statistically significant difference among treatments, 2-Methoxyphenol seemed to affect germination and cotyledon emergence to a greater extent, whereas 2,6-Di-*tert*-butyl-4-methylphenol had a greater impact on seedling size. The effect of the mixture was greater than that of both compounds separately.

## 1. Introduction

This is an age in the geological and biological history of Earth in which the signs of human action are evident all over the planet, and in this scenario, agriculture faces the challenge of feeding a population that is constantly growing. Traditional agricultural practices, such as the extensive use of herbicides, have been effective in terms of productivity; however, they have contributed significantly to the emergence of resistant weeds and environmental deterioration through the loss of biodiversity and soil contamination [1].

The proliferation of weeds that compete for nutrients reduces crop yield, affects agricultural infrastructure (e.g., water channelling systems), and hinders the harvest and commercialisation processes, which requires an active search for more natural alternatives to the use of synthetic herbicides.

Since antiquity, different studies have shown the harmful effects of certain plants on the growth of other plants, as well as their interaction with microorganisms. This interaction among plants mediated by secondary metabolites is known as allelopathy [2]. Allelopathy has gained interest in agricultural systems, with research focusing on the interference between crop plants and weeds to regulate the latter. Different practices are being implemented, such as allelopathic topsoil, cultural management and natural herbicides, with the aim of improving weed control. However, despite the fact that ecological management and restoration methods are improving as a result of exhaustive research on allelopathy, this phenomenon continues to be underused in modern agricultural practices [3].

A better understanding of the types of allelopathic substances produced by plants, as well as their purposes and mechanisms of action, offers promising solutions for weed management, crop protection and environmental conservation [4]. It would be ideal to obtain and employ natural, water-soluble compounds with potential herbicide properties at low concentrations, low molecular weight and simple formulae, which would facilitate their extraction or synthesis and application; in addition, they should be species-specific and biodegradable and present low toxicity, thereby being safer for humans, animals and the environment. Most phenolic compounds have several of these properties, which make them promising candidates for the development of bioherbicides. Phenolic acids are common secondary metabolites that can act as allelochemicals and suppress the germination and growth of competing plants in the soil [5]. Despite their wide variety and the evidence on the molecular mechanisms of many of them [6,7,8], most studies are focused on allelopathic compounds of extracts, with a lack of information about the phytotoxicity of many species [9]. Some of these compounds are volatile and can be concentrated by essential oil extraction. Some essential oils, such as those of *Cinnamomum zeylanicum*, *Thymus vulgaris*, *Eucalyptus globulus*, *Rosmarinus officinalis* and *Lavandula angustifolia*, have been proven to significantly inhibit the germination and growth of several species of common and problematic weeds [10,11,12]. These results suggest that essential oils may be as effective as some synthetic herbicides, with the additional benefit of being biodegradable and less toxic to the environment. Evaluating the activity of all phenolic compounds would allow finding compounds with new mechanisms of action and uncharted places of action that could lead to solving the problem of weed resistance to conventional herbicides [13]. On the other hand, it should be noted that the action of allelopathic plants is commonly caused by the mixture of several allelochemicals found in the soil at very low concentrations [9]. For this reason, it is always interesting to investigate whether there is an interaction that amplifies the effect of potential herbicidal compounds. The phytotoxicity of phenolic compounds may vary considerably depending on their chemical structure, the concentration applied, the presence of other compounds, and the species and substrate used [6,14,15,16,17,18,19,20].

*Cistus ladanifer* Linneo (1753) is a typical allelopathic species of the Mediterranean ecosystem. *Cistus ladanifer* secretes a large amount of exudate in its leaves and photosynthetic stems. This exudate is called labdanum and has properties highly appreciated in the perfume industry [21]. When the jaral tree is exploited, cutting it to extract its essential oil or labdanum, plant richness increases immediately in the area, although it decreases in time when the population of the labdanum tree regenerates [22]. Several studies indicate that this reduction of biodiversity is caused by the allelopathic potential of this species [14,23,24,25], whose essential oil also presents phytotoxic activity [26,27]. The labdanum and essential oil of its leaves contain a high concentration of phenolic compounds, some of which have not been studied to date [15,28,29,30,31,32,33,34]. The essential oil of *C. ladanifer* is a complex mixture of volatile compounds derived from secondary metabolism, with α-pinene, viridiflorol, pinocarveol, p-cymene, camphene, bornyl acetate and ledol being the majority [35]. The composition varies quantitatively depending on the different regions of origin of the plant material [36,37,38,39]. These variations are associated with environmental factors, geographical variations and genetic characteristics, giving rise to commercial products with heterogeneous and undefined qualities [40,41]. 2-methoxyphenol and 2,6-di-*tert*-butyl-4-methylphenol are minor compounds identified in oils from Spanish regions [25,42,43].

As in most cases, allelopathy derives from the joint effect of the mixture of multiple compounds in different proportions, with the minor compounds being equally important [17,28,29]. Therefore, it is important to evaluate the activity of each of the components independently of the quantity in which it is present.

In this context, the current study aims to contribute to understanding the allelopathic potential of *Cistus ladanifer* and to identifying possible structures that may serve as sources of safer and less polluting natural herbicides. After determining which phenolic compounds that are present in *Cistus ladanifer* essential oil are commercially available and have not been evaluated as possible strong herbicides, the objectives of this study were to (1) analyse the phytotoxic activity of 2-Methoxyphenol and 2,6-Di-*tert*-butyl-4-methylphenol against monocotyledons and dicotyledons; (2) determine whether the activity of these compounds is maintained when they are applied in soil; and (3) identify interactions among these compounds that may increase or reduce their effects. To this end, we conducted standardised bioassays with low concentrations (0.01 to 1 mM) of 2-Methoxyphenol and 2,6-Di-*tert*-butyl-4-methylphenol, separately and as a mixture, against seeds of *Lactuca sativa* and *Allium cepa*. The effects of these compounds were analysed in terms of germination, cotyledon emergence and seedling growth to determine their viability as natural herbicides. 2-Methoxyphenol, also known as Guaiacol (Figure 1A), exists naturally in guaiac wood resin and essential oil. It is used as a non-stick agent in surface coating, a flavouring agent, and an antioxidant (for fats, oils and vitamins), and with medicinal purposes as an expectorant, an antiseptic and a local anaesthetic [44]. On the other hand, 2,6-Di-*tert*-butyl-4-methylphenol or Butylhydroxytoluene (Figure 1B) is found in its natural form in the lychee fruit. It is used as a food additive (E-321) and an antioxidant in cosmetics, pharmaceuticals, aeroplane fuels, rubber, petroleum derivatives, electrical transformer oil, and embalming fluid [45].

## 2. Results

### 2.1. Effect of 2-Methoxyphenol and 2,6-Di-tert-butyl-4-methylphenol on the Germination of Lactuca sativa *L.*

In the tests conducted on paper, 2-Methoxyphenol, 2,6-Di-*tert*-butyl-4-methylphenol and the mixture of both compounds inhibited the total germination (%Gt) and germination rate (%GR) of *Lactuca sativa* at the four concentrations analysed, with this inhibition being up to 75% for the highest concentration (Figure 2). However, in the tests performed in soil, although %GR was inhibited with all treatments, %Gt was only inhibited when the compounds were applied as a mixture at 0.5 and 1 mM.

It is worth highlighting that the greatest effects were observed at the highest concentrations. The effect of the mixture of compounds on %Gt and %GR showed a significant positive linear correlation with concentration (R^2^ = 0.98 and R^2^ = 0.91, respectively), i.e., the higher the concentration, the greater the inhibition. The half-maximal inhibitory concentration (IC_50_) was 0.5 mM and 0.01 mM, respectively (Figure 2). Moreover, the inhibitory effect was greater when the compounds were applied as a mixture with respect to the effect of the separate compounds. This difference was statistically significant at 1 mM for %Gt, %GR on paper and at 0.5 and 1 mM for %Gt in soil.

It is also important to point out that, on paper, the effect of 2-Methoxyphenol at 1 mM on %GR was similar to that of the mixture and significantly greater than that of 2,6-Di-*tert*-butyl-4-methylphenol.

### 2.2. Effect of 2-Methoxyphenol and 2,6-Di-tert-butyl-4-methylphenol on the Cotyledon Emergence of Lactuca sativa *L.*

The analysed compounds presented the same behaviour on total cotyledon emergence (%Ct) and cotyledon emergence rate (%CR) as on %Gt and %GR, although inhibition was greater in all analysed concentrations both on paper and in soil. On paper, %Ct and %CR were strongly inhibited up to over 75% at the highest concentrations, whereas the effect was lower in soil; however, in the latter case, both the separate compounds and their mixture, in addition to inhibiting cotyledon emergence rate, also inhibited %Ct at all analysed concentrations. Furthermore, it was observed that, at any of the applied concentrations, 2-Methoxyphenol delayed cotyledon emergence to a significantly greater extent compared to 2,6-Di-*tert*-butyl-4-methylphenol (Figure 3).

Similarly, the greatest inhibitory effects were observed at the highest concentrations, and the effect of the mixture of compounds on %Ct and %CR showed a significant positive correlation with concentration (R^2^ = 0.97 and R^2^ = 0.99, respectively). IC_50_ was lower than 0.01 mM in both cases and when the mixture was applied at 1 mM in soil, the inhibitory effect on %Ct and %CR was significantly greater than that of the separate compounds (Figure 3).

### 2.3. Effect of 2-Methoxyphenol and 2,6-Di-tert-butyl-4-methylphenol on the Seedling Growth of Lactuca sativa *L.*

2-Methoxyphenol, 2,6-Di-*tert*-butyl-4-methylphenol and the mixture of both phenols also significantly inhibited the seedling growth of *Lactuca sativa* at all analysed concentrations in the paper tests. 2,6-Di-*tert*-butyl-4-methylphenol inhibited radicle length (%Radicle length) and hypocotyl length (%Hypocotyl length) to a greater extent than 2-Methoxyphenol when applied at 0.5 and 1 mM (Figure 4).

In soil, the effects of 2-Methoxyphenol and 2,6-Di-*tert*-butyl-4-methylphenol were low, with only 2-Methoxyphenol significantly inhibiting radicle and hypocotyl growth at 1 mM. Moreover, 2,6-Di-*tert*-butyl-4-methylphenol significantly stimulated hypocotyl growth at 0.01 and 0.1 mM. On its part, the mixture also stimulated hypocotyl growth at 0.1 mM, although it inhibited %Radicle length and %Hypocotyl length at 0.5 and 1 mM (Figure 4).

Both in soil and on paper, the greatest inhibitory effects on seedling growth were observed at the highest concentrations. On paper, the effect of the mixture of compounds on %Radicle length and %Hypocotyl length showed a significant logarithmic correlation with concentration (R^2^ = 0.97 and R^2^ = 0.91, respectively). IC_50_ was lower than 0.1 mM in both cases. It is also important to highlight that when the mixed compounds were applied at 0.5 mM in soil, radicle and hypocotyl growth was inhibited, whereas the separate compounds at this concentration did not show a significant difference from the control (Figure 4).

### 2.4. Effect of 2-Methoxyphenol and 2,6-Di-tert-butyl-4-methylphenol on the Germination of Allium cepa *L.*

The paper test with seeds of *Allium cepa* revealed that 2-Methoxyphenol and the mixture of the two compounds inhibited neither total germination (%Gt) nor germination rate (%GR) at any of the four analysed concentrations. Only 2,6-Di-*tert*-butyl-4-methylphenol at 1 mM produced significantly lower values of %Gt and %GR with respect to the control (Figure 5). The soil test showed that 2,6-Di-*tert*-butyl-4-methylphenol inhibited %Gt and %GR at low concentrations. In turn, as was expected, 2-Methoxyphenol and the mixture of both compounds inhibited %Gt and %GR at high concentrations (0.5 and 1 mM). It is worth pointing out that the mixture applied at 1 mM in soil inhibited %Gt, whereas the separate compounds at the same concentration did not have an inhibitory effect, and the mixture applied at 0.5 mM inhibited %GR, while the separate compounds at the same concentration did not have an inhibitory effect either (Figure 5).

### 2.5. Effect of 2-Methoxyphenol and 2,6-Di-tert-butyl-4-methylphenol on the Cotyledon Emergence of Allium cepa *L.*

On paper, 2-Methoxyphenol inhibited total cotyledon emergence (%Ct) from a concentration of 0.1 mM and cotyledon emergence rate (%CR) at all four analysed concentrations. In soil, although %Ct was only inhibited at 1 mM, the effect on %CR was 20% greater at all concentrations (Figure 6).

On paper, 2,6-Di-*tert*-butyl-4-methylphenol only inhibited %Ct at 1 mM without significantly affecting %Ct; however, in soil, this compound inhibited %Ct at 0.01 mM and %CR at all concentrations (Figure 6).

Similarly, the mixture of both phenolic compounds also presented a greater inhibitory effect in soil. On paper, the mixture inhibited %CR at 0.5 and 1 mM, whereas, in soil, it inhibited %CR at all analysed concentrations (Figure 6).

For all paper treatments, the greatest inhibitory effects were observed at the highest concentrations, which was not observed in any case in the soil tests.

### 2.6. Effect of 2-Methoxyphenol and 2,6-Di-tert-butyl-4-methylphenol on the Seedling Growth of Allium cepa *L.*

2-Methoxyphenol and 2,6-Di-*tert*-butyl-4-methylphenol and the mixture of the two phenols significantly inhibited the %Radicle length and %Hypocotyl length of the seedlings of *Allium cepa* at all analysed concentrations when applied both on paper and in soil, except for %Hypocotyl length at 0.1 and 0.5 mM when 2-Methoxyphenol was applied, %Radicle length at 0.1 and 0.5 mM when 2,6-Di-*tert*-butyl-4-methylphenol was applied in soil, and %Radicle length at 0.01 and 0.1 mM when the mixture was applied on paper (Figure 7).

It is important to underline that when the mixed compounds were applied in soil, radicle and cotyledon growth was inhibited to a greater extent than when the compounds were applied separately (Figure 7).

## 3. Discussion

The repeated and frequent use of non-selective herbicides, such as glyphosate, has led to environmental health problems and the appearance of resistant weeds. In the face of possible prohibitions by institutions that advocate for the change toward more sustainable practices and safer agriculture, it is necessary to search for more natural products with new mechanisms of action that decompose rapidly in soil, and that can be used in ecological agriculture.

Essential oils of allelopathic plants may contain allelochemicals that inhibit germination [46], apparently by damaging meristematic cells [47]. Essential oils are classified as “generally regarded as safe” (GRAS) and can be used as a viable technology for the control of weeds in organic agricultural systems, although it is necessary to obtain basic information about their compounds and the phytotoxic potential of each of these compounds. An important group to which the phytotoxic activity of essential oils has been attributed is constituted by phenolic compounds [46,47].

In this work, standardised bioassays are performed with low concentrations (0.01 to 1 mM) of 2-Methoxyphenol and 2,6-Di-*tert*-butyl-4-methylphenol, separately and as a mixture, against seeds of *Lactuca sativa* and *Allium cepa*. The results showed that these phenolic compounds present in *Cistus ladanifer* essential oil, as others that have been previously studied, presented phytotoxic activity to a greater or lesser extent depending on the substrate, the species and the concentration analysed [19,48,49].

For the tests conducted on paper with seeds of *Lactuca sativa*, 2-Methoxyphenol, 2,6-Di-*tert*-butyl-4-methylphenol and the mixture of both compounds not only delayed cotyledon germination and emergence significantly, but they also directly inhibited germination, cotyledon emergence and seedling growth at all concentrations analysed. The fact that these compounds showed up to over 50% inhibitory activity at very low concentrations (0.01 mM) suggests that they could be considered as candidates for possible sources of bioherbicides. In general, high concentrations of phenols tend to inhibit germination and seedling growth, whereas low doses may stimulate these processes through the phenomenon known as hormesis [50,51]. This study was focused on assays with very low concentrations, although the greatest inhibitory effects were obtained with the highest concentration (1 mM), and when the mixture was applied, there was a dependence on the dose in all indexes measured, detecting neither hormetic nor synergistic effects. Pearson’s correlation analysis showed that R^2^ > 0.95, calculating an IC_50_ of 0.5 mM for %Gt, 0.01 mM for %GR, %Radicle length and %Hypocotyl length, and less than 0.01 mM for %Ct and %CR. The fact that the effect depends on concentration is further evidence of the phytotoxic potential of phenolic compounds. This finding of allelochemicals with the characteristic of presenting phytotoxic activity at very low concentrations is significant for ensuring the effectiveness of the product and minimising the costs and environmental impact in a possible application. In the agricultural industry, new generations of herbicides are effective at very low doses, such as sulfonylureas, imidazolinones and triazolopyrimidines, which inhibit the synthesis of essential amino acids for growth at less than 30 g/ha [52]. Even so, proper management is critical, as low doses and high specificity can also lead to the selection of resistant weeds if used repeatedly and without rotation of action mechanisms.

Some species may be sensitive to certain compounds, whereas others may show greater resistance. When the assays were carried out on paper with seeds of *Allium cepa*, the effects of 2-Methoxyphenol, 2,6-Di-*tert*-butyl-4-methylphenol and the mixture of both were lower than those observed in *Lactuca sativa*. Against this monocotyledonous species, none of the treatments had a significant effect on germination or %GR. Cotyledon emergence and %CR were only inhibited at the highest concentrations, and although the different solutions significantly inhibited seedling growth, the effects were lower than in the assays with *Lactuca sativa*. This delay in cotyledon emergence and the inhibition of radicle and hypocotyl growth may be as detrimental to the establishment of the seedling as the inhibition of germination [53]. Phenolic acids increase the activity of phenylalanine ammonia-lyase (PAL) and β-glucosidase, whereas they also reduce the activity of phenol-β-glucosyltransferase, inhibiting radicle growth. They can also affect DNA and RNA integrity and inhibit protein synthesis [54]. Some phenolic acids, such as benzoic acid and cinnamic acid, reduce the content of chlorophyll, thereby inhibiting photosynthesis [55]. Other phenolic acids reduce the growth of receptive plants through the alteration of hormones [56,57,58], membranes [57,59,60,61] and respiration [60,62]. Guaiacol o 2-Methoxyphenol is commonly used as a substrate to measure the activity of peroxidases, enzymes involved in processes such as cell elongation and defence against oxidative stress. This suggests that its interaction with enzymatic systems could indirectly influence plant growth and development, although it depends on factors such as concentration and physiological context. Other studies show that phenolic compounds, such as chlorogenic acid, caffeic acid and catechol, inhibit germination by altering the functionality of λ-phosphorylase, which is a key enzyme in the process [2,60]. This enzyme is present in both monocotyledons and dicotyledons. Thus, it is deduced that, in this case, the mechanism of action of these compounds could be different, making them more specific to dicotyledons. Understanding these mechanisms is essential to optimise the application of these compounds in crop management [63]. This specificity could be used to develop more specific and effective weed control strategies [64]. One of the most widely used herbicides in agriculture due to its effectiveness in controlling broadleaf weeds (dicotyledons) in crops such as corn, wheat, rice, barley and sorghum is 2,4-dichlorophenoxyacetic acid. This compound is a phenolic derivative, with an oxygen atom attached to the benzene ring instead of an -OH group. This phenoxy group is crucial for its activity by imitating the action of auxins plant growth hormones, causing uncontrolled growth of leaves and stems and inhibiting root development, which ends up killing the plant [65].

The assays conducted with different substrates showed that the efficacy of phenols can be affected by environmental factors. When the treatments were carried out in a universal substrate, the inhibitory effects on %Gt were lower, and only the mixture of phenolic compounds significantly inhibited the germination of lettuce and onion seeds at 0.5 and 1 mM, respectively. Microbial degradation affects the final destination and concentration of allelochemical substances in the surrounding soil and their activity [66,67,68,69]. Isolated microorganisms from a hydroponic lettuce crop supplied with ferulic acid degraded this allelochemical compound, reducing its phytotoxic effects [70]. This would explain the lower effects observed when the analysed solutions were applied in the soil. The rest of the indexes measured in soil were also lower than on paper with *Lactuca sativa* seeds; however, a different behaviour was detected in *Allium cepa*, observing that, when any of the solutions were applied in this substrate, the inhibition of %CR in *A. cepa* was greater than on paper. The same occurred with %Radicle length and %Hypocotyl length; when the mixture was applied in soil, the inhibition of seedling growth in *A. cepa* was greater than on paper. Once released, allelochemicals accumulate in the soil [20] and endure the processes of the latter, such as retention, transport and transformation [71]. These processes may be different for each compound. Due to its physicochemical characteristics, 2-Methoxyphenol is expected to volatilise in humid soil and show very high mobility and low absorption; in the atmosphere, this phenol is expected to be susceptible to solar photolysis and to have a half-life of a few hours [44]. In turn, 2,6-Di-*tert*-butyl-4-methylphenol is expected to be immobile in soil and to degrade in a few days, as well as to be unstable in water and susceptible to photolysis [45]. Moreover, transformation or biodegradation may lead to other compounds that are more or less active with a recipient plant, which could explain the differences found among species. 2-Methoxyphenol can lose its methoxy group (-OCH_3_), thereby becoming a catechol (1,2-dihydroxybencene), which becomes 1,2-benzoquinone. In its decomposition, 2,6-Di-*tert*-butyl-4-methylphenol could lead to more simple phenolic compounds and benzoquinones. Catechol, as was previously mentioned, may alter the functioning of essential enzymes and proteins for photosynthesis and respiration by reacting with their functional groups [2,72]. Some quinones, such as sorghum and juglone, also affect plant growth, as they prevent photosynthesis and cell respiration by decoupling electron transport in mitochondria and chloroplasts [73]. Nowadays, there are herbicides based on these types of synthetic compounds. Thus, the differences between monocotyledonous and dicotyledonous in radicle structure, cell wall composition, metabolism and stress response lead to variations in the degree and type of damage of the assayed allelochemicals and their degradation products generated in the soil.

Comparing the activity between the analysed phenols, there was no clear difference in their behaviour, although we identified some significant differences between them. On paper, against both lettuce and onion, 2-Methoxyphenol had a significantly greater effect on %GR and %Ct, whereas 2,6-Di-*tert*-butyl-4-methylphenol inhibited radicle and hypocotyl length to a greater extent at 0.5 mM. In soil, 2-Methoxyphenol also showed a greater effect than 2,6-Di-*tert*-butyl-4-methylphenol on %Ct, %CR and seedling growth of *Lactuca sativa*, whereas in *Allium cepa*, 2,6-Di-*tert*-butyl-4-methylphenol presented a greater effect on the measured indexes. This confirms not only that the effects depend on the compounds and the concentration applied but also that their efficacy can be affected by different environmental factors that may influence chemical degradation or transformation, as well as the influence of microorganisms and the physicochemical properties of the substrate.

It is worth highlighting that when the mixture of the two compounds was applied at 1 mM in soil, the germination of *Lactuca sativa* and *Allium cepa* was inhibited, whereas their separate application at this concentration did not show significant differences with respect to the control. Furthermore, the %Ct, %CR, and hypocotyl length of lettuce were significantly lower with the mixture of phenols compared to the separate phenols. In addition, when the mixture was applied at 0.1 mM onion seeds, cotyledon emergence and radicle length were significantly inhibited, which was not observed with the separate phenols. This confirms the interaction of the compounds. Other studies of artificial mixtures such as this one reveal that the joint action of different phenolic acids shows synergistic or additive effects [74,75,76]. The allelopathic capacity of a species often results from the collective or synergistic response of several allelochemicals rather than from a single compound [17]. The mixture in equal proportions of 20 allelochemicals identified in *Vulpia myuros* showed synergistic and additive effects with respect to the effect that each of the compounds presented separately, but the effect was less than that obtained when the compounds were mixed in the proportions that these compounds were found in vulpia extract [77].

To sum up, 2-Methoxyphenol and 2,6-Di-*tert*-butyl-4-methylphenol presented phytotoxic properties that may interfere with the germination, cotyledon emergence and seedling growth of monocotyledonous and dicotyledonous plants (*Allium cepa* and *Lactuca sativa*). The tests conducted at different concentrations and in different substrates revealed that the efficacy of these phenols may be particularly affected by the conditions of the assay. The chemical formula of the compound could be key to its effectiveness against a target species, although this aspect may be less relevant among compounds of the same chemical group if soil degradation leads to the same products. Factors such as concentration can also be an essential aspect of the activity of the compound, which could range from inhibition to stimulation. Due to biodegradation processes, activity in soil is expected to be lower than on paper; however, the chemical transformation of some compounds may lead to other products that are more phytotoxic than the precursor. Another condition that must be considered is the combination of the different compounds that may be found in the substrate, which often results in greater phytotoxic activity compared to that of isolated compounds. Lastly, compounds can show different effects when they interact with different species. Making use of these characteristics, we can develop more selective and effective formulae at lower concentrations, thereby minimising the negative environmental impacts. The comparison of these mechanisms is crucial to optimise their possible application in crop management.

Therefore, it is concluded that, in the current situation, it is urgent to search for safe, natural compounds with herbicide properties that minimise the environmental impact when applied to crops in weed control. Essential oils of allelopathic plants show great potential as a source of natural herbicides [47]. Some of their components, such as phenolic compounds, present unique properties that make them especially appropriate candidates. This work not only contributes to understanding the herbicide activity of two phenolic compounds present in *Cistus ladanifer* essential oil, but it also offers guidelines on the methodology to follow in the study of these natural products.

## 4. Materials and Methods

This study proposes a bioassay that contemplates all these variables, using commercial seeds of *Lactuca sativa* (dicotyledonous) and *Allium cepa* (monocotyledonous), which are characterised by their uniform sensitivity and response capacity in phytotoxic tests [6,78,79]. Whatman filter paper and commercial substrate were employed to verify whether the activity of the compounds was affected by the physical-chemical and biological conditions of the soil. The compounds were applied separately and as a mixture in order to detect possible interactions at low concentrations (0.01 to 1 mM) and in a sufficiently wide range to identify possible hormetic effects [50,51].

### 4.1. Reagents, Seeds and Substrates

Reagents of 2-Methoxyphenol (CAS-No:90-05-1) and 2,6-Di-*tert*-butyl-4-methylphenol (CAS-No:128-37-0) for synthesis with a purity of >98% were acquired from Sigma-Aldrich to prepare the solutions of 0.01 mM, 0.1 mM, 0.5 mM and 0.1 mM of each of the compound separately. For the mixture of the two phenols, four solutions of these four concentrations were prepared with equimolar concentrations of each of the compounds. The pH varied between 6.1 and 6.3 from one solution to another. There were no significant differences in pH among solutions.

We acquired commercial seeds of *Lactuca sativa* L., romaine verte maraîchère variety (Vilmorin jardin—CS70110—38291 St Quentin Fallavier Cedex—France), as a representative of dicotyledons, and seeds of *Allium cepa* L., Bianca di Maggio variety (Vilmorin Jardin—CS70110—38291 St Quentin Fallavier Cedex—France), as a representative of monocotyledons. These species are ideal for phytotoxicity bioassays [79] and present a total germination of over 98% after being watered with distilled water.

The substrates used were Whatman No. 118 paper and substrate of universal type, based on 95% peat, 5% green compost, and 1.3 Kg/m^3^ fertiliser: 12N + 12P + 17K (Geolia, Aki Brico-lage España S.L. B-839857—Madrid, Spain). The commercial universal soil presented the following characteristics: organic matter per dry matter (60%), electrical conductivity (40 mS/m), apparent dry density (320 g/L), grain size (0–20 mm), and pH 5.5–6.5.

None of the materials (seeds and substrate) were sterilised before the experiment.

### 4.2. Phytotoxicity Bioassay

A standardised test was followed to evaluate the phytotoxic activity of the selected phenolic compounds [78,80]. In Petri dishes, 50 seeds of *Lactuca sativa* or *Allium cepa* were placed on paper or on 25 g of universal substrate, and they were watered with Milli-Q water for the controls and with different solutions for the treatments (5 mL for paper and 16 mL for soil). Four replicates for each treatment were sealed with Parafilm to prevent evaporation for 5 and 6 days for the seeds of *L. sativa* and *A. cepa*, respectively, in a germination chamber at 15 h of light and 9 h of darkness at a constant temperature of 22 °C.

### 4.3. Measured Indexes to Quantify the Phytotoxic Effect

Total germination (%Gt) and total cotyledon emergence (%Ct): On the last day of the experiment, the number of germinated seeds and the number of germinated seedlings with cotyledons were counted. With these data, we calculated the mean values of the four replicates, which are expressed as percentages relative to the control [19,48,49].

Germination rate (%GR) and cotyledon emergence rate (%CR): In order to measure a possible delay in germination and cotyledon emergence, we counted the number of germinated seeds and cotyledons that appeared every day, and the indexes were calculated using the following Equation (1):(1)$$S=\left(N_{1}\cdot 1\right)+\left(N_{2}-N_{1}\right)\cdot \frac{1}{2}+\left(N_{3}-N_{2}\right)\cdot \frac{1}{3}+\ldots \left(N_{n}-N_{n-1}\right)\cdot \frac{1}{n}$$
where N_1_, N_2_, N_3_, … N_n−1_, N_n_ are the proportions of germinated seeds or emerged cotyledons obtained in the first (1), second (2), third (3), …, (n − 1), (n) days [80]. The results were expressed as percentages relative to control.

Radicle growth (%Radicle length) and hypocotyl growth (%Hypocotyl length): To determine a possible alteration of growth, on the last day, 10 seedlings selected at random from each Petri dish were measured, and we calculated the mean values, which were expressed as a percentage relative to the control [81].

### 4.4. Statistical Analysis

The significance level of the comparisons among treatments was estimated using the Mann-Whitney U test. The differences were considered significant when *p* < 0.05. The interrelationships between germination and seed growth with the concentration of phenolic compounds were determined by Pearson’s determination coefficient. The effective concentrations required to induce half-maximal inhibition of growth (IC_50_) were calculated according to the linear or logarithmic relationship between concentration and per cent inhibition of plant growth. All statistical analyses were conducted using the statistical software IBM SPSS Statistics version 26.

## Figures and Tables

**Figure 1 plants-14-00022-f001:**
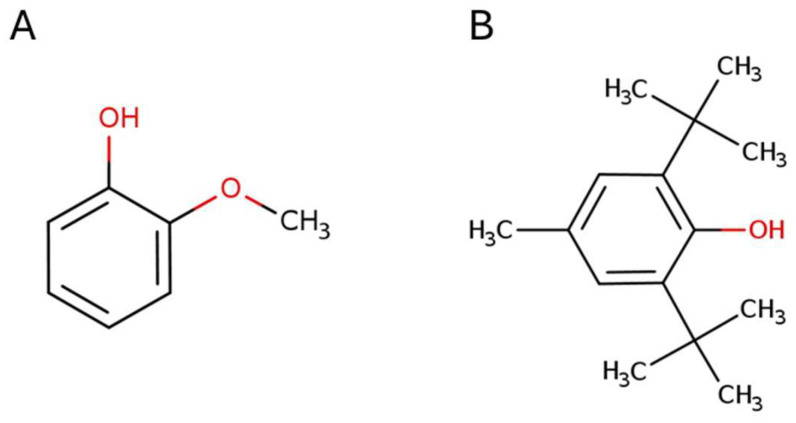
Chemical structure of 2-Methoxyphenol (**A**), 2,6-Di-*tert*-butyl-4-methylphenol (**B**). Source: https://echa.europa.eu/ (accessed on 28 October 2024).

**Figure 2 plants-14-00022-f002:**
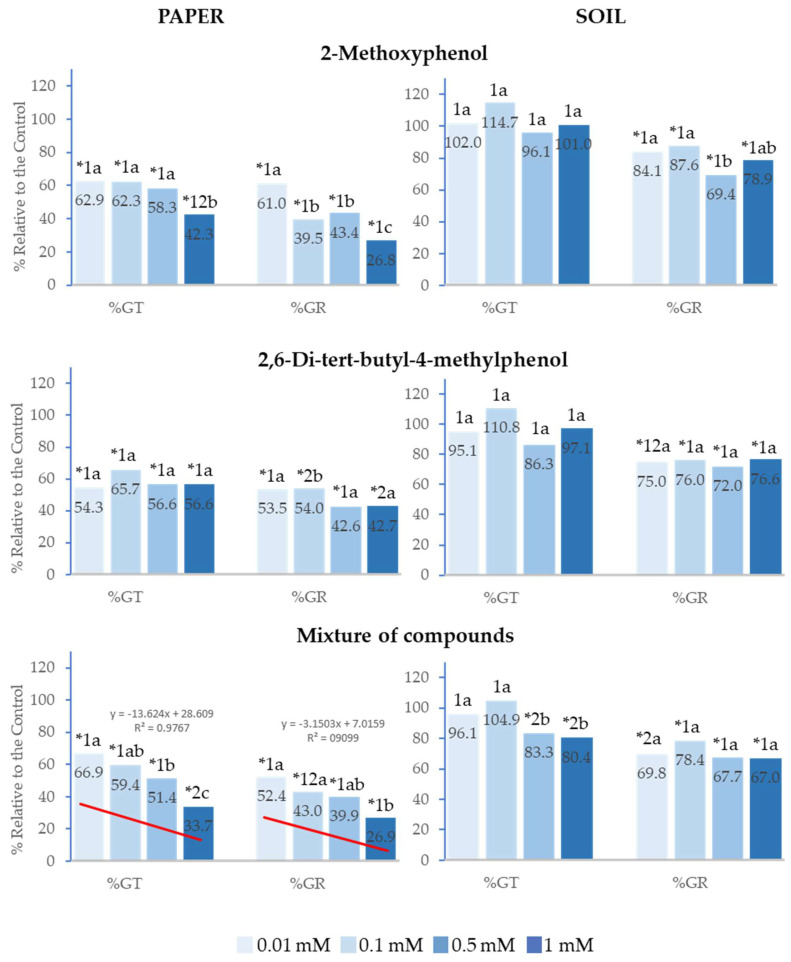
The effect of different concentrations of 2-Methoxyphenol and 2,6-Di-*tert*-butyl-4-methylphenol and the mixture of the two compounds on the total germination (%Gt) and germination rate (%GR) of *Lactuca sativa*, expressed as a percentage relative to the control. Four replicates of each treatment were performed (*n* = 4 × 50 = 200 seeds in total for each solution). * Significantly different from the controls; 1, 2 different numbers indicate significant differences between treatments of the same index and for each concentration. a, b, c: different letters indicate significant differences between concentrations of the same index and for each treatment. *p* < 0.05 (Mann–Whitney U test).

**Figure 3 plants-14-00022-f003:**
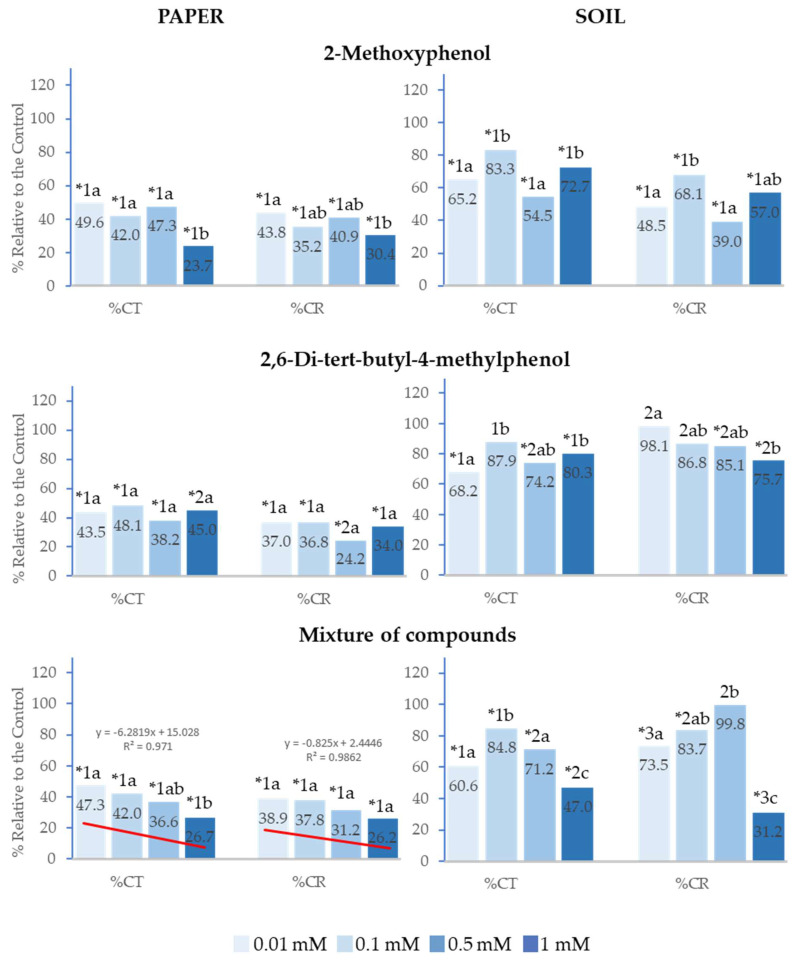
The effect of different concentrations of 2-Methoxyphenol and 2,6-Di-*tert*-butyl-4-methylphenol and the mixture of the two compounds on the total cotyledon emergence (%Ct) and cotyledon emergence rate (%CR) of *Lactuca sativa*, expressed as a percentage relative to the control. Four replicates of each treatment were performed (*n* = 4 × 50 = 200 seeds in total for each solution). * Significantly different from the controls; 1, 2, 3: different numbers indicate significant differences between treatments of the same index and for each concentration. a, b, c: different letters indicate significant differences between concentrations of the same index and for each treatment. *p* < 0.05 (Mann–Whitney U test).

**Figure 4 plants-14-00022-f004:**
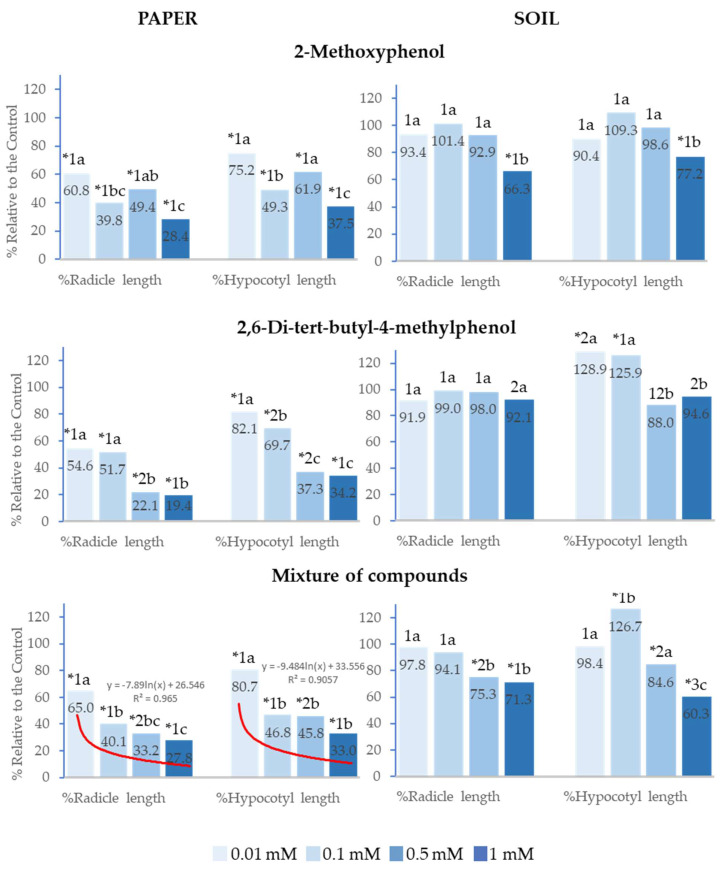
Effect of different concentrations of 2-Methoxyphenol and 2,6-Di-*tert*-butyl-4-methylphenol and the mixture of the two compounds on the radicle and hypocotyl length of *Lactuca sativa*, expressed as a percentage relative to the control. Four replicates of each treatment were performed (*n* = 4 × 50 = 200 seeds in total for each solution). * Significantly different from the controls. 1, 2, 3: different numbers indicate significant differences between treatments of the same index and for each concentration. a, b, c: different letters indicate significant differences between concentrations of the same index and for each treatment. *p* < 0.05 (Mann–Whitney U test).

**Figure 5 plants-14-00022-f005:**
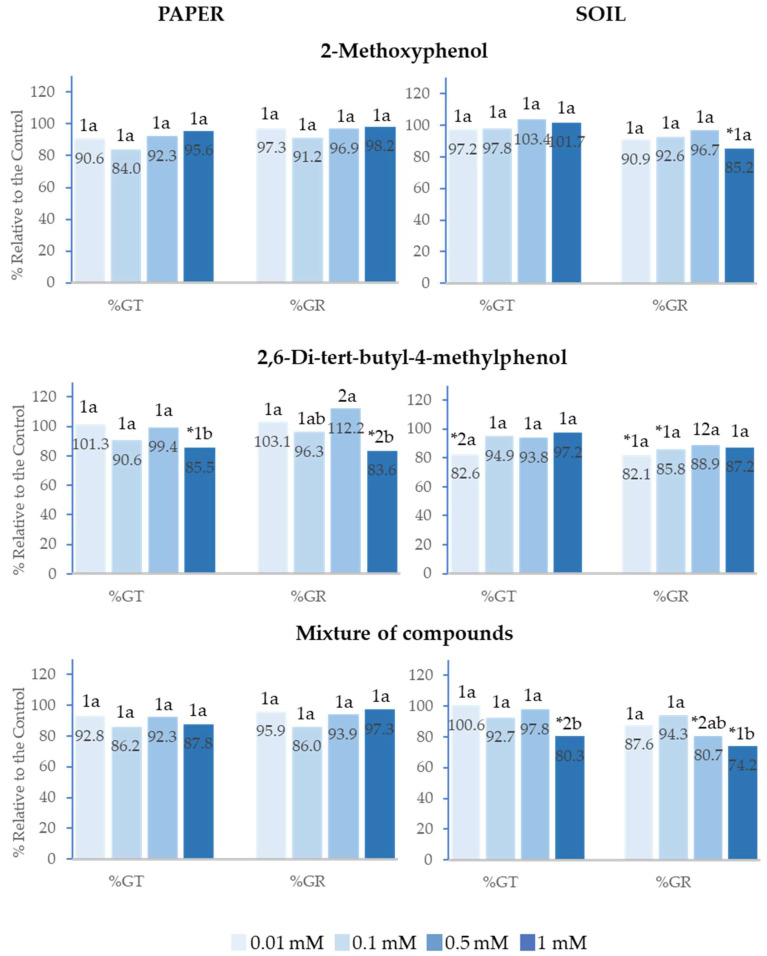
The effect of different concentrations of 2-Methoxyphenol and 2,6-Di-*tert*-butyl-4-methylphenol and the mixture of the two compounds on the total germination (%Gt) and germination rate (%GR) of *Allium cepa*, expressed as a percentage relative to the control. Four replicates of each treatment were performed (*n* = 4 × 50 = 200 seeds in total for each solution). * Significantly different from the controls; 1, 2 different numbers indicate significant differences between treatments of the same index and for each concentration. a, b: different letters indicate significant differences between concentrations of the same index and for each treatment. *p* < 0.05 (Mann–Whitney U test).

**Figure 6 plants-14-00022-f006:**
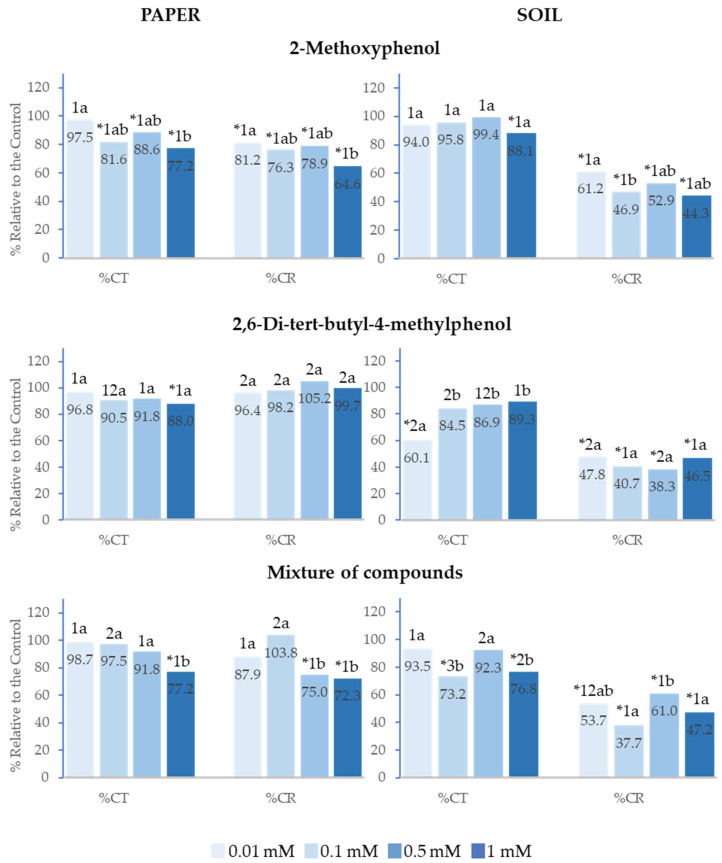
Effect of different concentrations of 2-Methoxyphenol and 2,6-Di-*tert*-butyl-4-methylphenol and the mixture of the two compounds on the total cotyledon emergence (%Ct) and cotyledon emergence rate (%CR) of *Allium cepa*, expressed as a percentage relative to the control. Four replicates of each treatment were performed (*n* = 4 × 50 = 200 seeds in total for each solution). * Significantly different from the controls. 1, 2, 3: different numbers indicate significant differences between treatments of the same index and for each concentration. a, b: different letters indicate significant differences between concentrations of the same index and for each treatment. *p* < 0.05 (Mann–Whitney U test).

**Figure 7 plants-14-00022-f007:**
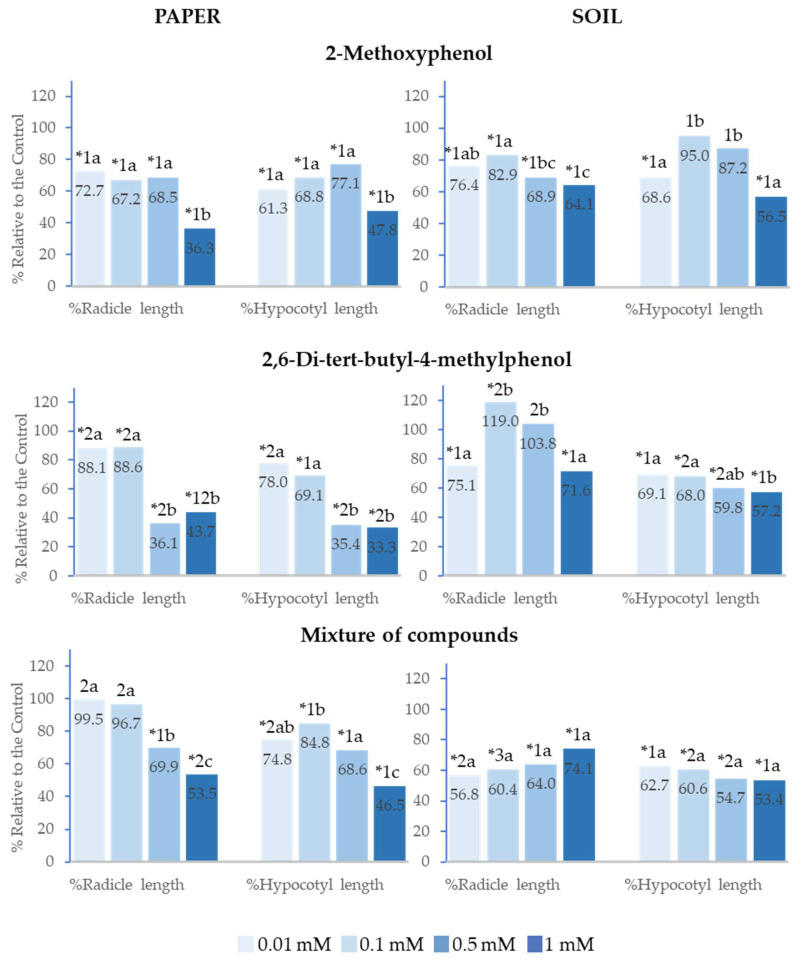
Effect of different concentrations of 2-Methoxyphenol and 2,6-Di-*tert*-butyl-4-methylphenol and the mixture of the two compounds on the radicle and hypocotyl length of *Allium cepa*, expressed as a percentage relative to the control. Four replicates of each treatment were performed (*n* = 4 × 50 = 200 seeds in total for each solution). * Significantly different from the controls. 1, 2, 3: different numbers indicate significant differences between treatments of the same index and for each concentration. a, b, c: different letters indicate significant differences between concentrations of the same index and for each treatment. *p* < 0.05 (Mann–Whitney U test).

## Data Availability

The data presented in this study are available on request from the corresponding author.

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
