# Peer review of "Phytotoxic Activity Analysis of 2-Methoxyphenol and 2,6-Di-tert-butyl-4-methylphenol Present in Cistus ladanifer L. Essential Oil"

_plants, 2024, doi:10.3390/plants14010022_

Round 1

Reviewer 1 Report

Comments and Suggestions for Authors

In this manuscript, Diego Orellana Dávila and colleagues have investigated two commercially available phenolic compounds of Cistus ladanifer essential oil (2-Methoxyphenol and 2,6-Di-tert-butyl-4-methylphenol) on the inhibition of germination, cotyledon emergence and seedling growth of Lactuca sativa for the tests conducted on paper. This study is interesting, and I have the following comments:

1, For the Title, I suggest to employ “Phytotoxic activity analysis of 2-Methoxyphenol and 2,6-Di-tert-butyl-4-methylphenol present in Cistus ladanifer L. essential oi”.

2, For the Abstract, the mixture ratio of 2-Methoxyphenol and 2,6-Di-tert-butyl-4-methylphenol should be described.

3, For the Keywords, “germination” and “seedling growth” could be included.

4, For the Introduction, previous studies on 2-Methoxyphenol and 2,6-Di-tert-butyl-4-methylphenol should be included.

5, For the Results, at least three different mixture ratios of 2-Methoxyphenol and 2,6-Di-tert-butyl-4-methylphenol should be examined in the revision.

6, For the Discussion, putative mechanism underlying Phytotoxic activity analysis of 2-Methoxyphenol and 2,6-Di-tert-butyl-4-methylphenol should be discussed in comparison with known bioherbicides.

7, For the Materials and methods, this section is too brief and necessary information is missing. For instance, commercial catalogs for 2-Methoxyphenol and 2,6-Di-tert-butyl-4-methylphenol should be provided. Genetic background of Lactuca sativa should be included. Growth condition like humidity, temperature, light cycle for the Lactuca sativa seedling should be described in the revision.

Author Response

Thank you very much for taking the time to review this manuscript. Please find the detailed responses below and the corresponding revisions/corrections in track changes in the re-submitted files.  

Comments 1: For the Title, I suggest to employ “Phytotoxic activity analysis of 2-Methoxyphenol and 2,6-Di-tert-butyl-4-methylphenol present in Cistus ladanifer L. essential oi”.  

Response 1: We agree with this comment. Therefore, we have made the change  

Comments 2: For the Abstract, the mixture ratio of 2-Methoxyphenol and 2,6-Di-tert-butyl-4-methylphenol should be described.  

Response 2: Agree. We have made the change  

Comments 3: For the Keywords, “germination” and “seedling growth” could be included.  

Response 3: We agree with this comment. Therefore, we have made the change  

Comments 4, For the Introduction, previous studies on 2-Methoxyphenol and 2,6-Di-tert-butyl-4-methylphenol should be included.  

Response 4: There are no previous studies on the phytotoxic activity of 2-Methoxyphenol and 2,6-Di-tert-butyl-4-methylphenol. The authors are the first to test these compounds.  

Comments 5, For the Results, at least three different mixture ratios of 2-Methoxyphenol and 2,6-Di-tert-butyl-4-methylphenol should be examined in the revision.  

Response 5: In our study, mixtures with different proportions of the compounds were not performed. Doing so now would require new controls that would not allow comparisons with the current results.  

Comments 6, For the Discussion, putative mechanism underlying Phytotoxic activity analysis of 2-Methoxyphenol and 2,6-Di-tert-butyl-4-methylphenol should be discussed in comparison with known bioherbicides.  

Response 6: Thank you for pointing this out. We agree with this comment. Therefore, we have added the requested information.  

Comments 7, For the Materials and methods, this section is too brief and necessary information is missing. For instance, commercial catalogs for 2-Methoxyphenol and 2,6-Di-tert-butyl-4-methylphenol should be provided. Genetic background of Lactuca sativa should be included. Growth condition like humidity, temperature, light cycle for the Lactuca sativa seedling should be described in the revision.  

Response 7: Agree. We have, accordingly, revised this point (line 463-473). The growth conditions are already described in the line 497-499. Regarding humidity we remember that each Petri dishes were sealed with Parafilm to prevent evaporation.

Reviewer 2 Report

Comments and Suggestions for Authors

The subject of the manuscript is very interesting and current because it concerns the potential use of natural substances as herbicides. In general, the manuscript is very well prepared, it brings new information about two compounds occurring in the plant Cistus ladanifer. The authors used chemically pure substances available on the market and a mix prepared by the authors. For Bioassay, they used frequently used representatives of dicotyledons and monocotyledons: Lactuca sativa and Allium cepa.

Critical remarks:

In the text of the  Introduction there is no information about the natural composition of essential oil of Cistus ladanifer, both qualitative and quantitative, so we do not have complete information if the title is correct as the authors used chemical substances bought in the market not obtained  from the plant. In addition, It is not clear whether the authors used the mix to imitate the natural share of these compounds. Is Cistus ladanifer the only species where both compounds occur together? The authors provide one example of a species where each of these compounds occurs. This list should be completed in the introduction (l. 85-90). Similarly, this aspect should be developed in the discussion. Are there any papers that examined artificial mixes of allelopathic compounds? These will be interesting issue for the discussion.

Text structure:

The introduction should be corrected in its final part: l. 80-108. Here, the aims of the work, which are presented in the discussion (sic!), should be clearly presented. Such detailed aims should be removed from the discussion (l. 279-286). In the introduction, l. 101-109 are actually a summary of methods and analyses – this should be removed in this form from this place. Unfortunately, in the journal formula, methods are at the end, so that is probably why the authors attempted such a summary in the introduction. Similarly, in the discussion, the authors too often repeat results that should be in the results chapter, e.g. l. 302-304. Such places should be rephrased.

Other:

l.346 – italic for A. cepa

Figures – the y-axis in the figures should be described, even if the unit is given (%)

Author Response

Thank you very much for taking the time to review this manuscript. Please find the detailed responses below and the corresponding revisions/corrections in track changes in the re-submitted files.

Comments 1: Critical remarks: In the text of the  Introduction there is no information about the natural composition of essential oil of Cistus ladanifer, both qualitative and quantitative, so we do not have complete information if the title is correct as the authors used chemical substances bought in the market not obtained  from the plant. In addition, It is not clear whether the authors used the mix to imitate the natural share of these compounds. Is Cistus ladanifer the only species where both compounds occur together? The authors provide one example of a species where each of these compounds occurs. This list should be completed in the introduction (l. 85-90). Similarly, this aspect should be developed in the discussion. Are there any papers that examined artificial mixes of allelopathic compounds? These will be interesting issue for the discussion.

Response 1: Thank you for pointing this out. We agree with this comment. Therefore, we have added the requested information.    

Comments 2: Text structure: The introduction should be corrected in its final part: l. 80-108. Here, the aims of the work, which are presented in the discussion (sic!), should be clearly presented. Such detailed aims should be removed from the discussion (l. 279-286). In the introduction, l. 101-109 are actually a summary of methods and analyses – this should be removed in this form from this place. Unfortunately, in the journal formula, methods are at the end, so that is probably why the authors attempted such a summary in the introduction. Similarly, in the discussion, the authors too often repeat results that should be in the results chapter, e.g. l. 302-304. Such places should be rephrased.

Response 2: We agree with this comment. Therefore, we have made the change.  

Comments 3: Other: l.346 – italic for A. cepa Figures – the y-axis in the figures should be described, even if the unit is given (%).

Response 3: Agree. We have, accordingly, modified this point.

Round 2

Reviewer 1 Report

Comments and Suggestions for Authors

Authors have postively respond to my comments in the revision.

Author Response

The authors reiterate our gratitude to the reviewer for the time dedicated
